# Microbial Fuel Cells as Effective Tools for Energy Recovery and Antibiotic Detection in Water and Food

**DOI:** 10.3390/mi14122137

**Published:** 2023-11-22

**Authors:** Giulia Massaglia, Giacomo Spisni, Candido F. Pirri, Marzia Quaglio

**Affiliations:** 1Department of Applied Science and Technology (DISAT), Politecnico di Torino, Corso Duca degli Abruzzi 24, 10129 Torino, Italy; giacomo.spisni@iit.it (G.S.); fabrizio.pirri@polito.it (C.F.P.); 2Center for Sustainable Future Technologies@Polito, Istituto Italiano di Tecnologia, Environment Park, Building B2 Via Livorno 60, 10144 Torino, Italy

**Keywords:** microbial fuel cells, energy recovery, biosensors, bio-electrochemical sensors, antibiotic contamination

## Abstract

This work demonstrates that microbial fuel cells (MFCs), optimized for energy recovery, can be used as an effective tool to detect antibiotics in water-based environments. In MFCs, electroactive biofilms function as biocatalysts by converting the chemical energy of organic matter, which serves as the fuel, into electrical energy. The efficiency of the conversion process can be significantly affected by the presence of contaminants that act as toxicants to the biofilm. The present work demonstrates that MFCs can successfully detect antibiotic residues in water and water-based electrolytes containing complex carbon sources that may be associated with the food industry. Specifically, honey was selected as a model fuel to test the effectiveness of MFCs in detecting antibiotic contamination, and tetracycline was used as a reference antibiotic within this study. The results show that MFCs not only efficiently detect the presence of tetracycline in both acetate and honey-based electrolytes but also recover the same performance after each exposure cycle, proving to be a very robust and reliable technology for both biosensing and energy recovery.

## 1. Introduction

Foods of animal sources, such as eggs, milk, honey, and meat, play an important role in our daily diet, and there has been a rapid increase in their consumption in recent years, as they have proved so beneficial for human health and well-being. Although their consumption has been shown to be very beneficial to human health and well-being, the extensive use of antibiotics on farms is generating increasing attention [1,2,3]. Indeed, substances such as hormones, antibiotics, and growth promoters are widely used to prevent and treat diseases in animals, stimulate their growth, and improve their nutrition [1,2,3]. Unfortunately, many of these substances can be transferred as contaminants into food products [3]. In this scenario, the European Food Safety Authority (EFSA), in line with the World Health Organization (WHO), has listed the presence of veterinary drug residues in food as a major risk factor for public health [3]. One of the greatest concerns is regulating and controlling food safety, especially monitoring the presence of veterinary medicines in animal-source foods [1,2,3]. Clear evidence has been provided of the adverse consequences for human health of antimicrobial resistance developed due to continued exposure to nonhuman antibiotics [4,5,6]. Antibiotic resistance is a global sanitary emergence: the greater the amount of antibiotic substances used, the greater the likelihood that bacteria will develop resistance mechanisms, reducing the effectiveness of antibiotic therapies in medicine. This growing awareness of the need and urgency to actively protect antibiotics from overuse [7,8] is driving the development of rules of administration and limits of use for antibiotics, as well as methods to detect veterinary drug residues in food of animal origin. The development of reliable, inexpensive, and easy-to-use detection methods is a major challenge that needs to be overcome to ensure a good level of food safety for the majority of the world’s population [9,10]. Such methods must effectively determine whether the amount of antibiotics is below the permissible maximum residue limits (MRL), they must make detections continuously and in situ, and finally, they must be able to be used even in complex matrices, such as food and biological samples [11,12,13,14,15,16].

In this regard, many different analytical methods, such as high-performance liquid chromatography (HPLC) coupled with mass spectrometry (MS) and liquid chromatography combined with tandem mass spectrometry (LC-MS/MS), are proposed in the literature to detect antibiotics accurately and simultaneously in different environmental media [12,13]. Other determination techniques, such as capillary electrophoresis (CE) [14], Raman Spectroscopy [15], and enzyme-linked immunosorbent assays [16], have been exploited to monitor drug residues in food products. Although very innovative, many of the biological and conventional methods cannot be implemented for routine on-site use because they require pretreatment, are expensive, and are insufficiently sensitive. With the main goal of overcoming all these limitations, biosensors have attracted great interest as effective methods for screening antibiotic residues. Biosensors not only offer high sensitivity and selectivity but also provide a high degree of automation, combined with cost efficiency, real-time measurements, and high throughput [17,18,19]. Basically, in the general scheme of biosensors, a biological recognition element interacts with the target compound, inducing a biological response that the physical transducer transforms in a detectable signal proportional to the content of the analytes [20,21,22,23]. Several biological recognition elements have been used in the design of biosensors, including cofactors, enzymes, antibodies, organelles, tissues, cells, and whole microorganisms [24]. Among these biological elements, microorganisms offer a viable alternative for fabricating biosensors due to their ease of manipulation, improved viability and stability in vitro, and ability to enhance biosensor performance [25]. However, further improvements are needed to develop more effective biosensors that can prove to be promising tools for the detection and quantification of antibiotics in food products [11]. One of the most promising types of bio-electrochemical sensors, based on whole microorganisms, are microbial fuel cells (MFCs) that convert chemical energy contained in fuel-acting organic compounds into electrical energy through the metabolic activity of so-called exoelectrogenic bacteria [26,27].

Since the output power is closely related to the metabolic activity of these microorganisms, the presence of a toxic substance that may interfere with their metabolism can directly affect the performance of the device [28,29,30,31,32]. Therefore, the steps of energy conversion and signal transduction are intimately coupled, avoiding the need for external transducers and additional power units. MFC-based biosensors can be successfully used as fast-response and low-maintenance detectors, being also inexpensive as they can be built from low-cost carbon-based materials [33,34]. Given their versatility of use, the application of MFCs for bioremediation of agro-food pollutants [35,36,37] has received great interest in recent decades by treating several wastewaters such as wine [38] and swine [39] and also focusing attention on the detection of trace of antibiotics in foods of animal origin [40]. Works are already available in the literature showing the ability of MFCs not only to survive exposure to antibiotics but also to be able to effectively keep producing energy, even during the exposure event [41].

In this work, air-cathode single chamber MFCs (a-SCMFCs) are proposed as bioelectrochemical systems for antibiotic detection, specifically choosing tetracycline as a model drug. The impact of the exposure of MFCs to tetracycline has been discussed by Li S. et al. [42]. In that work, the ability of electroactive biofilm to adapt and change composition in the presence of tetracycline has already been discussed in detail. The minimum concentration of tetracycline analyzed in that work is 10 mg/L.

In particular, this study aims to demonstrate the effectiveness of MFCs as a tool to detect the presence of antibiotics in small amounts in honey, chosen as a model animal food. Honey dissolved in water will serve the function of a carbon source for the sustenance of anodic microorganisms in MFCS. Antibiotic contamination in honey is a significant problem given its wide use in the food industry. This makes the choice of honey as a model system of particular relevance. It is important to consider that the European Union, as well as the USA and Japan, which are the main honey consumers, have not set specific limitations for antibiotics in honey [43]. Indeed, the maximum allowable residue limit (MRL) for tetracycline in honey for human consumption has been set at 20 μg/kg by the European Union as the recommended concentration for screening, but being a recommendation, the actual concentration can be higher [43]. In light of these data, to demonstrate the robustness of MFCs to detect tetracycline, a concentration of (3.53 ± 0.13) μg/kg was used as the target, more than 6 times lower than the EU-recommended MRL.

Experiments were performed testing the effect of tetracycline dissolved in two different water-based solutions serving as electrolytes for two separate sets of MFCs. The first electrolyte was obtained by dissolving honey in the water-based solution, thus using honey as the carbon source for sustaining the anodic microorganisms [44]. The second electrolyte was a reference water-based electrolyte solution containing sodium acetate as the carbon energy source. The target amount of tetracycline was dissolved in both electrolyte solutions during the experiments, creating several cycles of exposure to antibiotics. At the same time, the experiment performed shows the recovery ability of SCMFCs. In fact, after exposure to tetracycline, all cells recovered their functioning, returning to produce the same current density measured without the drug. All the results obtained demonstrate the effectiveness of SCMFCs as tools to detect low concentrations of tetracycline in honey simply by diluting it in water and using it as a carbon power source.

Finally, the behavior of all devices is also analyzed by evaluating the amount of energy recovered (E_rec_) per unit volume of the electrolyte [45,46,47,48]. E_rec_ values obtained from both electrolytes with and without antibiotics were compared, demonstrating the ability of microorganisms not only to accurately detect the presence of tetracycline but also to allow energy recovery under all test conditions. This ability to generate energy by recovering functionality after antibiotic exposure demonstrates the enormous potential of MFCs as low-energy detection systems.

## 2. Materials and Methods

### 2.1. Materials and Nanofiber Synthesis

With the aim to improve biofilm growth, composite nanostructured anodes were fabricated [44]. They were obtained by decorating carbon paper (CP, from Fuel Cell Earth, Woburn, MA, USA) with nanofibers (NFs) made of polyethylene oxide (PEO, Sigma Aldrich, Darmstadt, Germany) with a molecular weight of 600 kDa. NFs were fabricated by directly electrospinning PEO-based solution on CP, which was used as a conductive support during the entire electrospinning process (NANON 01A equipment, from MECC Co., Ltd., Fukuoka, Japan). This process ensures the formation of neatly distributed PEO-NFs on the CP surface, with preferential alignment along the conducting structures protruding from the CP surface [44]. The introduction of PEO-NFs on the surface of CP allows the surface area to be increased, thus promoting proliferation of microorganisms. In addition, NFs composed of PEO swell in water-based electrolyte, acting as a biomass carrier that promotes biofilm adhesion.

### 2.2. MFC Architecture and Configuration

In the present work, SCMFCs were fabricated via 3D printing (Object 3D, Al.Tip srl, Torino, Italy) with a squared single-chamber architecture and with an open-air cathode [35]. Membrane-free a-SCMFCs were selected, with the electrolyte shared between the anode and cathode compartments. The inner volume of the cells was 12.5 mL, and both anodes and cathodes showed a geometrical surface area close to 5.76 cm^2^. The structure of the a-SCMFCs used in this work is shown in Figure 1. CP/PEO-NFs were employed as composite anodes. Commercial CP electrodes (Fuel Cell Earth, Woburn, MA, USA) decorated with polytetrafluoroethylene (PTFE) on the outer side, the one in contact with air, and by a layer of Pt/C-based catalyst on the inner side, in contact with the electrolyte, were chosen as cathodes [42]. Electrons were collected with titanium wires (Goodfellow Cambridge Ltd., Huntingdon, UK) connected by a carbon cement (Leit-C, Sigma Aldrich, Darmstadt, Germany) to the electrodes.

The water-based solutions used for the electrolytes were prepared as discussed in a previous article [44]. Both the carbon sources, i.e., sodium acetate (C_2_H_3_NaO_2_) and honey, were added to the solutions at a concentration of 2 g/L. Minerals and nitrogen sources necessary to sustain microorganisms were introduced by adding ammonium chloride (NH_4_Cl) and potassium chloride (KCl) at concentrations of 0.31 g/L and 0.13 g/L, respectively. To regulate pH at a neutral value, sodium dihydrogen phosphate (NaH_2_PO_4_, 2.450 g/L) was also added. All the reagents were purchased from Sigma Aldrich (Darmstadt, Germany).

Each electrolyte under analysis was tested in duplicate. A mixed consortium from seawater natural sediment was employed. All the devices were run in fed-batch mode, replacing the electrolyte once the voltage output dropped at a value close to 0 V. A data-acquisition unit (Keysight 34970A, Agilent Technologies, Santa Clara, CA, USA) was used to monitor the performance of a-SCMFCs.

To test SCMFCs as bio-electrochemical sensors for antibiotics, a small quantity of tetracycline was added to both the electrolytes at a concentration of (3.53 ± 0.13) μg/kg, with respect to the amount of the carbon source. The selected value of tetracycline was lower than the MRL recommended for honey as a food matrix, which is close to 20 μg/kg. The experiments were run with an external load of 1 kΩ applied to each a-SCMFC. The ability of a-SCMFCs to recover their performance after exposure to toxic antibiotics was also investigated. To this purpose, the energy recovery (E_rec_) factor [45,46,47] was analyzed for both the electrolytes (sodium acetate and honey), with and without tetracycline. The following formula was used: $E_{rec}=(\int Pdt)/Vin$. The E_rec_ is indeed an energy density (J m^−3^) calculated as the ratio of the overall energy output estimated by (∫Pdt) and the inner volume of the reactor $V_{in}$. Electrochemical Impedance Spectroscopy (EIS) measurements were then carried out to investigate the internal resistance of all a-SCMFCs. A sinusoidal wave with amplitude of 25 mV was used as the signal, with the frequency ranging between 150 and 200 mHz.

## 3. Results and Discussion

### 3.1. a-SCMFCs for Tetracycline Detection in Solium Acetate-Based Electrolyte

In MFC-based biosensors, the biofilm acts as the sensing element. Thanks to its metabolism, it is directly responsible for the conversion of the chemical signal associated with the fuel and analytes available in the electrolyte into the output electrical signal [33]. Thus, in MFC-based biosensors, the steps of energy conversion and signal transduction are intimately coupled. Strategies that are effective in promoting energy conversion are thus expected to be useful in optimizing biosensing as well. For this reason, in this work, CP/PEO-NF-based nanostructured anodes enhance bacterial proliferation and biofilm adhesion to the anode, with the aim of optimizing the response of SCMFCs as biosensors. In all the experiments, an external load of 1 kΩ was used to investigate the performance of a-SCMFCs with both the electrolytes with and without antibiotics. In this way, since the geometric area of the electrodes of the a-SCMFCs used for the experiment is also equal, the trends of current density referring to all the tests performed were comparable.

In the first experiment, a water-based electrolyte with sodium acetate as the carbon energy source was used as a reference, and subsequently, tetracycline was added in a concentration equal to (3.53 ± 0.13) μg⁄kg. In Figure 2, the values of current density obtained during the experiment are reported. The first peak refers to the current density produced when only sodium acetate was present in the electrolyte. The following peaks show the results obtained during exposure to the antibiotic and during the subsequent recovery phase. An abrupt decrease in current density of about 4 times the value before exposure to the antibiotic was observed. This result demonstrates the strong impact of tetracycline on the metabolic activity of microorganisms.

Since the intensity of current density measured during exposure to antibiotics is so reduced because of the exposure to the toxicant, the peaks referring to the event are magnified and highlighted in the yellow box of Figure 2. When only sodium acetate is dissolved in the water-based electrolyte, the maximum current density value was (24.8 ± 0.1) mA/m^2^, while the maximum current density value obtained during tetracycline exposure was (5.87 ± 0.13) mA/m^2^, a full four times lower. Moreover, it is possible to appreciate that the presence of tetracycline contaminating the electrolyte also affects the duration of the peaks, which were shorter than when only sodium acetate was used. These results are in line with the findings of Li et al. [42], confirming the robustness of MFCs when exposed to antibiotics, but clearly shows that the MFC-based biosensors can produce an unambiguous signal in response to antibiotics amounts significantly lower than those tested in previous studies [41].

### 3.2. a-SCMFCs for Tetracycline Detection in Honey-Based Matrix

The analysis was further extended by investigating a-SCMFCs as bio-electrochemical sensors for tetracycline when dissolved in honey-based electrolyte. Results from these tests are fundamental to exploring the possibility of using these devices for the direct detection of drug residues in food matrices without the necessity to extract antibiotics before their detection, thus overcoming several limitations correlated to analytical methods reported in the literature [8,9,10,11,12,13,14,15,16]. Figure 3 shows the performance of a-SCMFCs in terms of the current densities achieved when the honey-based electrolyte was used and when tetracycline was added directly into this food matrix. Devices using honey-based electrolytes produced a maximum current density of (16.29 ± 1.02) mA/m^2^, which is one order of magnitude higher than the peak value of (0.65 ± 0.16) mA/m^2^ obtained in the presence of tetracycline into the electrolyte, as reported in Figure 3. An overall 95% decrease in current density was thus obtained after exposure to tetracycline, with a reduction in the duration of the peaks. These results show a strong similarity between the behavior of the cells running on sodium acetate and those fed with honey. Thus, the comparison of Figure 2 and Figure 3 immediately clarifies that the reduction in the duration of the current density peaks is independent of the specific carbon source but is only related to the presence of tetracycline that interferes with the metabolic activity of microorganisms.

These latter results allow us to demonstrate the ability of electroactive biofilms based on mixed consortia to adapt their metabolism to use complex substrates such as honey not only to produce electricity in MFC devices but to sense the presence of small quantities of tetracycline dissolved inside due to the strong impact it has on their metabolic activity.

Finally, the comparison of Figure 2 and Figure 3 allows us to confirm the capability of a-SCMFCS to recover the output current density after exposure to antibiotics for both sodium acetate-based and honey-based electrolytes. This result clearly demonstrates the robustness of the electroactive mixed consortia to survive toxic events, in line with the findings of other works [41,42]. Indeed, the electroactive biofilm was not only able to detect a very low amount of tetracycline dissolved in water but simultaneously to preserve the metabolic activity despite the contact with residues of drugs. The results obtained for honey-based electrolytes demonstrate that the robustness of MFCs is not compromised by the use of complex matrices.

### 3.3. Energy Recovery Analysis and Electrochemical Impedance Spectroscopy Results

The impact of tetracycline on the duration of current peaks can be better analyzed and discussed by introducing the recovered energy factor. E_rec_ is obtained by calculating the output energy produced by the a-SCMFC during an operational phase, i.e., the time between two refills of the electrolyte. This time frame is the same one during which the current density peak is observed. The values of E_rec_ calculated for the previously discussed experiments are shown in Figure 4 for both the two electrolytes with and without tetracycline.

Figure 4a refers to the results obtained for sodium acetate-based electrolyte. It is possible to appreciate that the starting value of E_rec_ of (252.4 ± 10.3) mJ/m^3^ was completely recovered after toxicant events, demonstrating once more the capability of microorganisms to recover their metabolic activity, producing the same electrical energy output [41,42]. At the same time, Figure 4a confirms the decrease in the electrical performance of the devices when exposed to tetracycline, which is reflected in the reduction in the value of E_rec_, close to (134.7 ± 8.5) mJ/m^3^. 

A similar trend can be observed by analyzing Figure 4b, which shows the E_rec_ values calculated when honey is used as the carbon source. In this case, the maximum value of E_rec_ is (326.2 ± 8.7) mJ/m^3^ and refers to the output energy produced before exposure to tetracycline. Exposure to tetracycline dissolved in the food-derived matrix caused a significant drop in the E_rec_ down to a value of (34.2 ± 4.3) mJ/m^3^. E_rec_ is thus reduced by almost an order of magnitude when honey-based electrolyte contains the antibiotic, also demonstrating, in this case, the capability of microorganisms to act as an efficient sensitive element. Even during this experiment, the electrical energy output is recovered after the toxicant events up to a value of (262 ± 9.8) mJ/m^3^. It is important to observe that the recovery is not complete. Indeed, the efficiency of the catabolic activity of the biofilm to transform honey is partially reduced by the exposure to tetracycline, as demonstrated by the reduction in E_rec_. Nevertheless, the minimum value of E_rec_ is still higher than the one reached during exposure to tetracycline by more than 6 times and pretty close to the values obtained for devices fed with sodium acetate. These latter results confirm that a-SCMFCs can be used as bio-electrochemical sensors for the detection of antibiotics from food-derived matrices without extraction and pretreatment.

Impedance behavior was investigated by performing EIS characterizations, which allowed us to provide an evaluation of internal resistance through the analysis of the charge transfer resistance [48,49]. The results are proposed in Figure 5.

In particular, EIS measurements were carried out on all a-SCMFCs both before and after exposure to the toxicant events. By this approach, it has been possible to investigate whether exposure to tetracycline could affect the internal resistance of a-SCMFCs after the sensing event. Figure 5a represents the Nyquist plots for a-SMCFCs running on sodium acetate-based electrolytes before and after the toxicant event. It is possible to appreciate that the total impedance values reached after the tetracycline’s detection results are close to the one obtained with only sodium acetate-based electrolyte (996 Ω and 1027 Ω, respectively). In Figure 5b, it can be observed that SCMFCs, initially fed with honey-based electrolyte, after exposure to the toxicant event, were characterized by total internal impedance values fully comparable with the one achieved when only honey is used as the fuel (991 Ω and 972 Ω, respectively).

These results are in line with all the other findings reported up to now and demonstrate that exposure to antibiotics has no detrimental effect on the interface between the biofilm and the electrode. Indeed, EIS measurements show that, for both the electrolytes, the charge transfer resistance at the interface is not changed by the exposure to tetracycline.

## 4. Conclusions

In the present work, we demonstrated that a-SCMFCs can be proposed as an effective tool for the detection of low amounts of tetracycline, close to (3.53 ± 0.13) µg/kg, a value significantly lower than the MRL recommended by the EU for honey as a food matrix (i.e., 20 µg/kg). Moreover, this work confirms the capability of these bio-electrochemical devices to detect antibiotic traces without the need to perform a purification and extraction process of the drug from the matrix before the analytical analysis. All obtained results confirm the pivotal role of electroactive biofilms made of microorganisms from mixed consortia that are able not only to convert complex food matrix, such as honey, into electrical energy but, at the same time, to sense the low amount of tetracycline dissolved in the fuel made of honey. A decrease in current density of 95% was achieved when tetracycline residues were added directly into the food-derived matrix, while a current density reduction of only 25% was detected by adding antibiotics into the reference electrolyte. Moreover, the maximum values of E_rec_ achieved after the toxicant event with sodium acetate were close to (254 ± 10.3) mJ/m^3^, quite similar to the value reached when honey is the fuel (262 ± 9.8) mJ/m^3^. At the same time, it is important to stress that during the exposure to tetracycline, an E_rec_ as low as (34.2 ± 4.3) mJ/m^3^ was obtained for devices operated with honey-based electrolyte, and this value is of one order of magnitude lower than the one reached when uncontaminated honey was employed. Finally, the electrical power output was recovered after toxicant events by devices fed with both sodium acetate and honey-based electrolytes. By demonstrating the feasibility of MFC-based sensors for detecting antibiotics in food matrices, the results of this study open up exciting new prospects for use, both by making electrolytes more complex and by defining the performance of this promising class of biosensors in greater detail.

## Figures and Tables

**Figure 1 micromachines-14-02137-f001:**
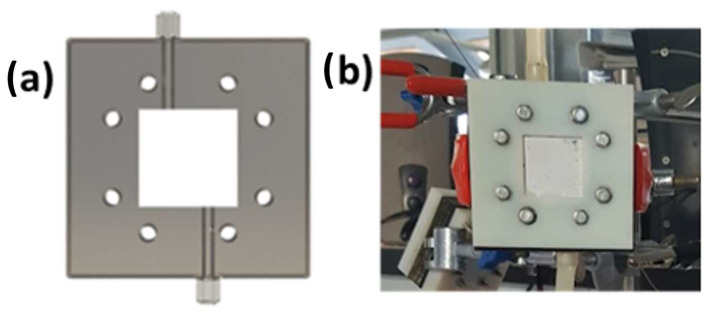
The structure of the a-SCMFC is proposed in (**a**) by a render that highlights the inner structure of the single chamber and in (**b**) by a picture of an actual cell.

**Figure 2 micromachines-14-02137-f002:**
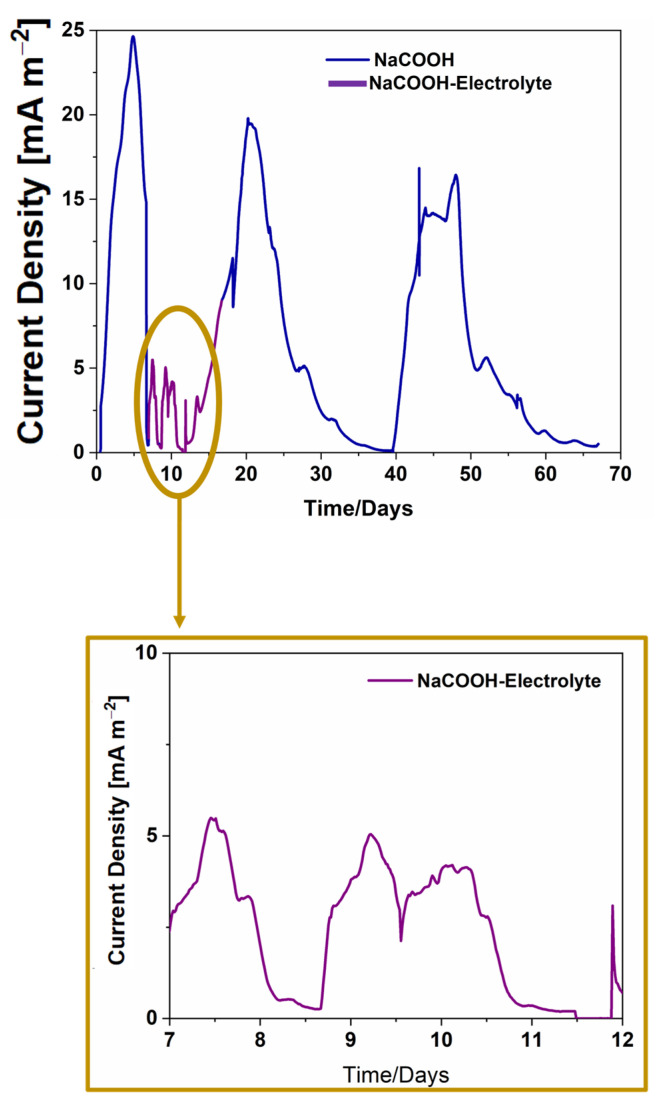
Current density production over time: the blue line refers to current density produced by a-SCMFCs running with sodium acetate, and the purple line shows the current density obtained during exposure to tetracycline. Results collected during exposure to antibiotics are magnified and highlighted in the yellow box.

**Figure 3 micromachines-14-02137-f003:**
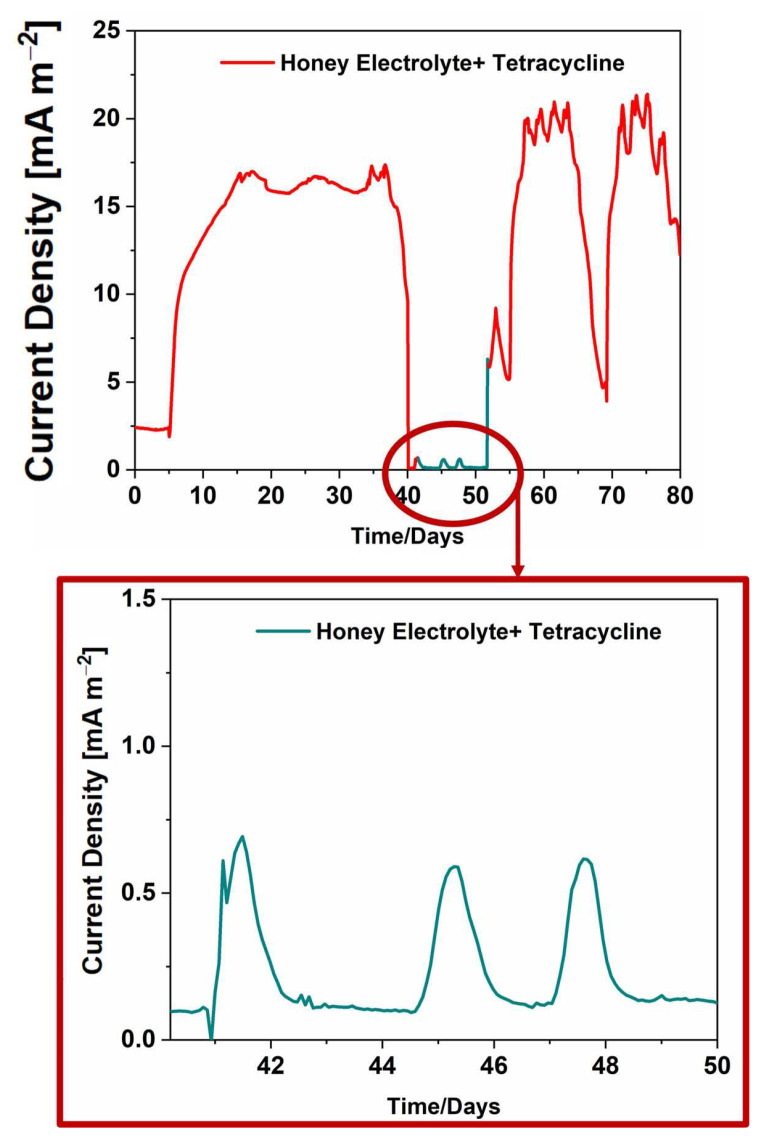
The red curve in the main graph refers to the current density for MFCs fed with honey, while in light blue, the current density produced during exposure to tetracycline is proposed. Current density during exposure to tetracycline is magnified and highlighted in the red box.

**Figure 4 micromachines-14-02137-f004:**
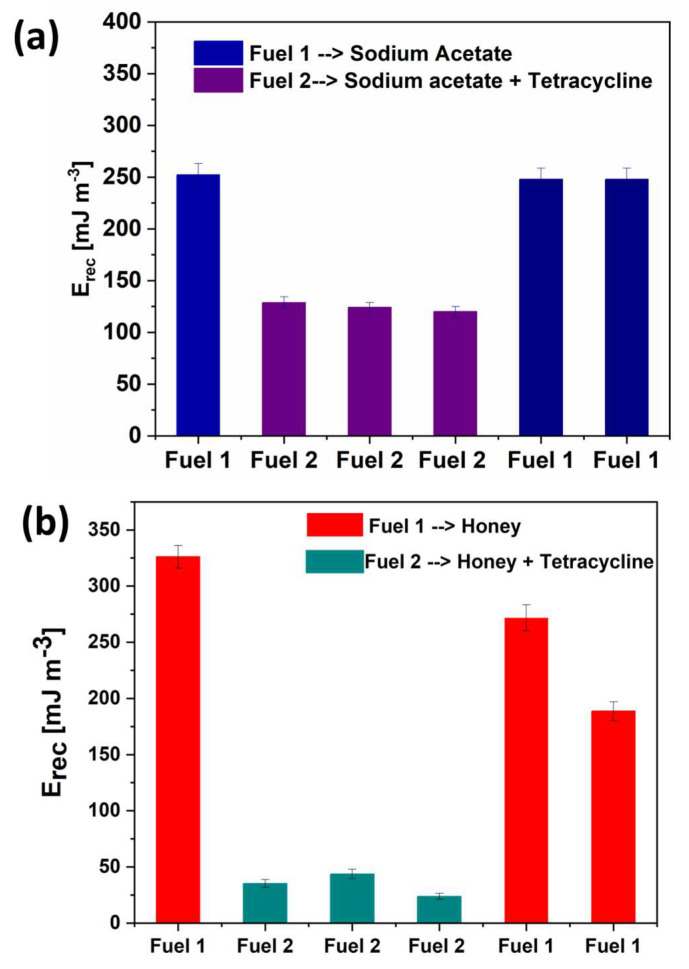
Recovered energy (E_rec_) calculated for SCMFCs using two different electrolytes: (**a**) the water-based electrolyte containing sodium acetate only and sodium acetate with tetracycline; (**b**) honey-based electrolyte and honey with tetracycline added. In both (**a**,**b**), Fuel 1 refers to the uncontaminated electrolyte, while Fuel 2 refers to the electrolyte with tetracycline.

**Figure 5 micromachines-14-02137-f005:**
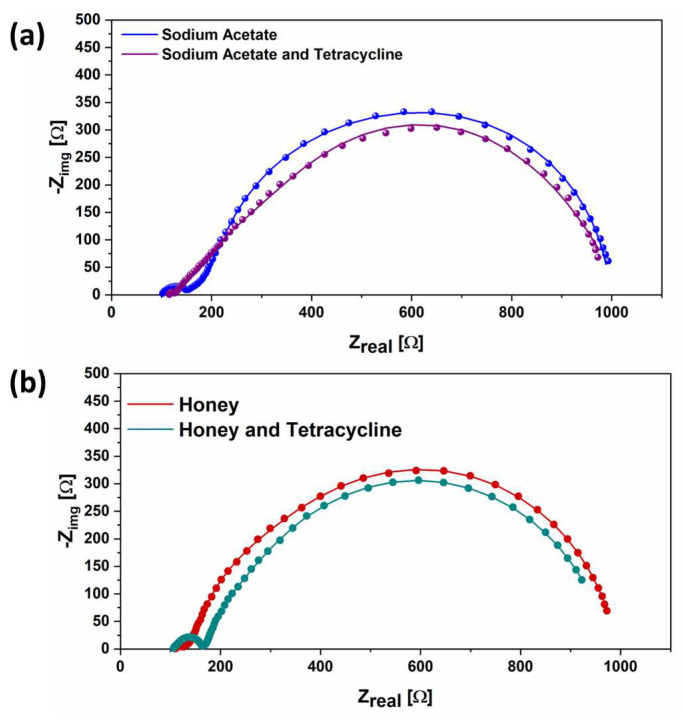
(**a**) Impedance spectra of SCMFCs when sodium acetate with and without tetracycline was used as the electrolyte. Sodium acetate (blue dots and line) and sodium acetate with tetracycline (pink dots and line). (**b**) Impedance spectra of SCMFCs when honey with and without tetracycline was used as electrolyte. Honey (dark red dots and line) and honey with tetracycline (dark cyan dots and line).

## Data Availability

Data are contained within the article.

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
