# Peer review of "Microbial Fuel Cells as Effective Tools for Energy Recovery and Antibiotic Detection in Water and Food"

_micromachines, 2023, doi:10.3390/mi14122137_

Round 1

Reviewer 1 Report

Comments and Suggestions for Authors

Typological errors can be rectified.

grammatical mistakes needs to clarified .

Authors detected low amount of tetracycline with honey as electrolyte.

But they have not explored about sensitivity and detection time.

Minimum and maximum limit of tetracycline concentration has to be calculated

proof reading of entire manuscript needed

Comments on the Quality of English Language

proof reading of entire manuscript needed

Author Response

Dear Dr Wenwei,

we would like to thank you and the Reviewers for the thorough evaluation and interest in the paper. Given below are the answers to the specific questions raised by the Reviewers and responses to their suggestions.

All the changes made in the revised version of the paper are highlighted in red in the response and in the manuscript.

We hope that the paper could now be suitable for publication.

Sincerely,

Prof. Marzia Quaglio and Dr. Giulia Massaglia on behalf of all Authors

Review Report - Reviewer 1

Comments and Suggestions for Authors

Typological errors can be rectified.

grammatical mistakes needs to clarified .

We have changed the text in the main manuscript in order to explain much more clearly the goals set, the results obtained and the conclusions that can be reached. These changes were also made taking care to correct grammatical errors. We hope that these changes will make the entire work clearer.

Authors detected low amount of tetracycline with honey as electrolyte. But they have not explored about sensitivity and detection time. Minimum and maximum limit of tetracycline concentration has to be calculated

With the main target to demonstrate the effectiveness of MFCs as tools to detect antibiotics into food matrix of animal origin, we selected an amount of tetracycline lower than the maximum detection limit (MRL) proposed as a recommended value by the EU, as better clarify in the work. The MRL is equal to 20 μg⁄kg while the selected value to prove feasibility of MFC-based sensors for tetracycline is about 3.5 μg⁄kg. Therefore, the main objective of the present work was not to demonstrate in detail MFCs as biosensors for the detection of tetracycline, but rather the capability to take over the presence of antibiotics also in complex matrix, without the need to extract the antibiotic from the matrix itself and only then verify the amount present.

For all above mentioned reasons, we did not explore different concentrations of tetracycline to define the sensitivity and the detection time, but rather preferred to investigate performance recovery and energy production.  This work has successfully demonstrated the capability of MFCs to detect the minimum value of tetracycline into honey selected as food matrix from animal origins.

proof reading of entire manuscript needed

Several changes and modifications have been made in order to ensure better readability of the entire manuscript.

Reviewer 2 Report

Comments and Suggestions for Authors

The paper was poorly written.

1.       The last paragraph of the Introduction should be put in the Result and Discussion section.

2.       The representations of Fig. 1, 2, and 3 are difficult to understand.

3.       The conclusion should be shorter and more concise.

4.       The configuration and structure of the MFC have not been shown in the paper.

5.       The author did not discuss how their MFC could be used in real-life situations.

Comments on the Quality of English Language

No comments on the quality of English language

Author Response

Dear Dr Wenwei,

we would like to thank you and the Reviewers for the thorough evaluation and interest in the paper. Given below are the answers to the specific questions raised by the Reviewers and responses to their suggestions.

All the changes made in the revised version of the paper are highlighted in red in the response and in the manuscript.

We hope that the paper could now be suitable for publication.

Sincerely,

Prof. Marzia Quaglio and Dr. Giulia Massaglia on behalf of all Authors

Review Report - Reviewer 2

Comments and Suggestions for Authors

The paper was poorly written.

We have changed the text in the main manuscript in order to explain much more clearly the goals set, the results obtained and the conclusions that can be reached. These changes were also made taking care to correct grammatical errors. We hope that these changes will make the entire work clearer.

  1. The last paragraph of the Introduction should be put in the Result and Discussion section.

We would like thank the Reviewer for this suggestion. The main manuscript was modified accordingly.

  1. The representations of Fig. 1, 2, and 3 are difficult to understand.

We appreciate the Reviewer comment and thus modify the Figure 1, 2, 3, so as to not only make the representation of the figures clearer but also to better highlight what was intended to be demonstrated.

In the revised version of the manuscript a new Figure 1 has been added as proposed by the Reviewer, so that the old Figure 1, 2, 3 are now numbered Figure 2, 3, 4 in the revised manuscript.

Figure 2 demonstrate the effectiveness of SCMFCs as tool to detect the presence of a low amount of tetracycline inside the water-based electrolyte. As shown in Figure 2, a decrease of current density of about 4 times can be observed when antibiotic traces were added to the electrolyte, demonstrating the impact of the tetracycline on the metabolic activity of microorganisms. At the same time, we extended the analysis investigating a-SCMFCs as bio-electrochemical sensors for tetracycline contaminating honey, selected as a model food of animal origin. An overall 95% of decrease is thus obtained for current density after exposure to tetracycline, with a reduction of the duration of the peaks, as discussed in Figure 3.

Since the presence of tetracycline in the two different electrolytes not only induces a decrease in the peak current reached by the devices, but also influences the duration of the peak itself, the impact of the antibiotic on the duration of the peak can be better analyzed by introducing the recovered energy factor in Figure 4. The correlation between current variation and peak duration in the presence or absence of the antibiotic allowed us to modify Figures 2 and 3 so that the characteristic peaks obtained in presence of antibiotic were represented with the same color used for representing the value reached by energy recovery. This was implemented for both the electrolytes analyzed.

Hopefully, this change will make Figures 2, 3 and 4 clearer and more interpretable.

  1. The conclusion should be shorter and more concise.

The conclusion was re-written to make it shorter and more concise.

  1. The configuration and structure of the MFC have not been shown in the paper.

We would like to thank the Reviewer for this consideration that allow us to better explain the structure and configuration of our devices, to this purpose a new Figure 1 has been added in the revised manuscript.

  1. The author did not discuss how their MFC could be used in real-life situations.

The main target of this work was to demonstrate the effectiveness of MFCs as tools to detect antibiotics in food matrix of animal origin. To reach this goal we selected tetracycline as the model antibiotic and honey as the model food. The reason for demonstrating the suitability of MFCs to be used as biosensors to detect the presence of antibiotics in food-derived matrix in related to the possibility to introduce the matrix in the electrolyte, without the need to extract the antibiotic from the matrix itself and only then verify the amount present.

We have changed the text in the main manuscript in order to explain much more clearly the goals set, the results obtained and the conclusions that can be reached.

Reviewer 3 Report

Comments and Suggestions for Authors

The choice of toxic compound is appropriate as the tetracycline often can be found in municipal and animal growth industry. Regarding the organic matter fuel, besides the acetic acid and the honey used, the authors could think for organic matter constituents typical for a specific wastewater; for example swine waste stream. In such a case a medium level of COD(BOD),  predominant organic matter composition and wastewater salinity can be accepted as a background of the matrix.     

Author Response

Dear Dr Wenwei,

we would like to thank you and the Reviewers for the thorough evaluation and interest in the paper. Given below are the answers to the specific questions raised by the Reviewers and responses to their suggestions.

All the changes made in the revised version of the paper are highlighted in red in the response and in the manuscript.

We hope that the paper could now be suitable for publication.

Sincerely,

Prof. Marzia Quaglio and Dr. Giulia Massaglia on behalf of all Authors

Review Report - Reviewer 3

Comments and Suggestions for Authors

The choice of toxic compound is appropriate as the tetracycline often can be found in municipal and animal growth industry. Regarding the organic matter fuel, besides the acetic acid and the honey used, the authors could think for organic matter constituents typical for a specific wastewater; for example swine waste stream. In such a case a medium level of COD(BOD),  predominant organic matter composition and wastewater salinity can be accepted as a background of the matrix.     

We would like to thank the Reviewer for this comment to allow us better explaining our approach. The main target of this work was to demonstrate the effectiveness of MFCs as tools to detect antibiotics in food matrix of animal origin. To reach this goal we selected tetracycline as the model antibiotic and honey as the model food. The reason for demonstrating the suitability of MFCs to be used as biosensors to detect the presence of antibiotics in food-derived matrix in related to the possibility to introduce the matrix in the electrolyte, without the need to extract the antibiotic from the matrix itself and only then verify the amount present.

In this work, indeed, we would like to demonstrate the effectiveness of MFCs as tools to preliminary detect the presence of antibiotics directly embedded into the food matrix from animal origins to meet the requirements defined by the World Health Organisation (WHO). In this context, WHO has identified antibiotic resistance as one of the three biggest threats to human health. Clear evidence has been provided of the adverse consequences for human health of the resistance to antimicrobials developed by organisms because of the continuous exposure to antibiotics for non-human use. For this reason, our target is not the wastewater treatment obtained by using the waste as carbon energy sources to couple the energy production and the water treatment. Rather, the main objective is to demonstrate the effectiveness of MFCs applied as a tool capable of detecting traces of antibiotic in even complex matrices, such as those derived from food. For this purpose, a low tetracycline concentration of (3.53 ±0.13) μg⁄kg was added to two different water-based electrolyte solutions. The first water-based electrolyte contained sodium acetate as carbon energy sources, and was used as a control within this work and thus it was directly compared to the target electrolyte, which was honey-based. The aforementioned concentration of tetracycline was selected being more almost 6 times lower than the maximum detection limit (MRL) of 10 μg⁄kg recommended for honey. Moreover, the same amount of tetracycline was dissolved in both electrolytes, leading thus to demonstrate the SCMFCs capability to detect the presence of drugs directly into the food-matrix without the necessity to extract the antibiotics before their detection. Exposure to antibiotics causes a sudden decrease in current density values, but after exposure SCMFCs recover their performance. All obtained results demonstrate the effectiveness of SCMFCs as tools to detect low concentration of tetracycline in honey. We also decided to analyze the amount of energy recovered (Erec) by a unit volume of the electrolyte [36-38]. The Erec values obtained for both electrolytes, honey-based and sodium acetate-based with and without antibiotics, was compared, demonstrating the ability of microorganisms to accurately detect the presence of antibiotics, guaranteeing also their recovery when the antibiotic is not present into electrolytes.

In any case to clarify and identify all possible applications of MFCs in the large field of sensors, we introduce the possibility to employ MFCs for waste treatments, evaluating organic matter constituents typical for a specific wastewater, for example swine waste stream. We hope that the contextualization of our work results to be clearer.

Reviewer 4 Report

Comments and Suggestions for Authors

Authors here attempt to present here the MFC can be a biosensor but the experimental design and data reported seems not supporting to their claim.

The introduction part is too vague. If aiming to glorify MFC in biosensors studies, please enrich the discusson on it avoiding general and well-known information. The last para which gives an overview of current studies is too large and presented as summary of work in extended for rather the key findings from this study.

Methodology section lacks crucial information like size and material of anode/cathode the amount of nanocomposite doped/coated on it, like ___mg/cm2, inocula and culture, medium composition, etc.

Figure 1, sodium acetate is CH3COONa, please correct it. Sodium chlorlode/honey can be the carbon cource for the electroactive bacteria not the electrolyte. There must be salts and buffers in the solution for electrolytes. please correct the presentation.

Section 3.1 and 3.2 comparision and discussion would be a better presentation rather individual observation reports.

Figure 4, EIS  discussion has to be improved if the authors think this is an interesting observation from this study. https://doi.org/10.1016/j.enzmictec.2019.01.007 is a good article on EIS and electrode performance in MFC.

Please avoid irrelevant references and discussion.

Overall improvement in the presentation is highly recommended for considering this study for further consideration.

Comments on the Quality of English Language

English is fine, a proofrad and gramatical corrections to be considered.

Author Response

Dear Dr Wenwei,

we would like to thank you and the Reviewers for the thorough evaluation and interest in the paper. Given below are the answers to the specific questions raised by the Reviewers and responses to their suggestions.

All the changes made in the revised version of the paper are highlighted in red in the response and in the manuscript.

We hope that the paper could now be suitable for publication.

Sincerely,

Prof. Marzia Quaglio and Dr. Giulia Massaglia on behalf of all Authors

Review Report - Reviewer 4

Comments and Suggestions for Authors

Authors here attempt to present here the MFC can be a biosensor but the experimental design and data reported seems not supporting to their claim.

The introduction part is too vague. If aiming to glorify MFC in biosensors studies, please enrich the discusson on it avoiding general and well-known information. The last para which gives an overview of current studies is too large and presented as summary of work in extended for rather the key findings from this study.

Methodology section lacks crucial information like size and material of anode/cathode the amount of nanocomposite doped/coated on it, like ___mg/cm2, inocula and culture, medium composition, etc.

We thank the Reviewer for this consideration. We decided to provide only crucial information in the Materials and Methods section highlighting that missing info have been extensively discussed in our previous work [ref. 44 in the revised manuscript].

Nevertheless, we agree that the article must be complete, for this reason we added the missing information in the revised manuscript.

Figure 1, sodium acetate is CH3COONa, please correct it. Sodium chlorlode/honey can be the carbon cource for the electroactive bacteria not the electrolyte. There must be salts and buffers in the solution for electrolytes. please correct the presentation.

We modify accordingly the Figure and its caption.         

Section 3.1 and 3.2 comparision and discussion would be a better presentation rather individual observation reports.

We modified the two section, providing comparison to other works and extending the discussion of the results.

Figure 4, EIS  discussion has to be improved if the authors think this is an interesting observation from this study. https://doi.org/10.1016/j.enzmictec.2019.01.007 is a good article on EIS and electrode performance in MFC.

With the main aim to be compliant with the auditor's advice, we improve the description and discussion of EIS Results. The proposed work investigates the effect of pH variation onto the performance of carbon xerogel doped with iron and nitrogen (CXFeNGO) and applied as efficient catalyst for cathode electrode. The authors proposed Shewanella oneidensis as pure culture pure culture from which to generate biofilm formation on the anode electrode. Subsequently, the authors induce variation of pH in the range from neutral pH and alkaline PH, leading thus to confirm the improved performance of CXFeNGO in alkaline pH, due to the increased of the medium. although there are many differences from our proposed work, since, for example, mixed culture is proposed for biofilm formation, the pH value of the electrolyte is kept neutral throughout the experiment due to the presence of the salts within the electrolyte, we introduce the reference into the main manuscript.

Please avoid irrelevant references and discussion.

We modify the main manuscript accordingly

Overall improvement in the presentation is highly recommended for considering this study for further consideration.

We have changed the text in the main manuscript in order to explain much more clearly the goals set, the results obtained and the conclusions that can be reached. We hope that the revised manuscript may now be suitable for publication.

Round 2

Reviewer 2 Report

Comments and Suggestions for Authors

The paper has been improved.

No further issue has been found in this revised version.